# [Re] Explaining Temporal Graph Models through an Explorer-Navigator Framework

**Miklós Hamar**                                        *miklos.hamar@student.uva.nl*
*University of Amsterdam*

**Matey Krastev**                                       *matey.krastev@student.uva.nl*
*University of Amsterdam*

**Kristiyan Hristov**                                   *kristiyan.hristov@student.uva.nl*
*University of Amsterdam*

**David Beglou**                                        *david.beglou@student.uva.nl*
*University of Amsterdam*

**Reviewed on OpenReview:** *https://openreview.net/forum?id=FI1XvwpchC*

## Abstract

Temporal graphs model complex dynamic relations that change over time, and are being used in a growing number of applications. Recently, several graph neural networks (GNNs) were proposed, designed specifically for this temporal setting (Xu et al., 2020; Rossi et al., 2020). However, these models are notoriously hard to interpret. For this reason, the original authors (Xia et al., 2023) propose the Temporal GNN Explainer (T-GNNExplainer) – an explorer-navigator framework to efficiently compute sparse explanations of target Temporal GNNs. We reproduce the main findings of the original paper, extend their work by proposing a different type of navigator method, and examine in detail its explanation capabilities and efficiency within various model and hyperparameter settings. We confirm that their explainer outperforms the other baselines across nearly all datasets and metrics. Our findings suggest the navigator helps bias the search process and that T-GNNExplainer can find an exact influential event set. Moreover, we examine the effect of different navigator methods and quantify the runtime-fidelity trade-off controlled by two hyper-parameters.

## 1 Introduction

Graphs and complex relational networks are well-known and studied tools for modelling an enormous variety of natural processes, being employed in nearly every domain. Similarly, Temporal Graphs are a specific subset of graphs designed to model complex dependencies within data, that may evolve across different time steps (Kostakos, 2009). Within Temporal Graphs, nodes and edges between data points can dynamically change with time, which makes them particularly useful for complex applications such as traffic prediction and forecasting, action identification, and anomaly detection, among others (Zhao et al., 2020; Cai et al., 2021; Yan et al., 2018).

The greater the importance of the applications of such models, the greater the need for more insight into their workings, which ensures that the decisions of the model are fair, unbiased and logically sound. In particular, Graph Neural Networks (GNNs), have been notoriously hard to interpret (Yuan et al., 2023). Therefore, various methods for post-hoc explanations of the decisions made by such graphs have been developed instead. However, many of those techniques fail to trivially apply to temporal graphs, as the latter are relatively novel models. To explain why a Temporal Graph Neural Network (T-GNN) made a certain prediction, one might want to find the subset of events preceding the current prediction, which has the highest influence.

Preferably, these explanations should also be sparse, in other words, only a small number of past events should be selected, because it helps interpretability.

To compute such explanations, Xia et al. (2023) propose the T-GNNExplainer method, which performs Monte Carlo Tree Search (MCTS) guided by an inductively trained navigator to efficiently compute sparse and accurate explanations. They demonstrate the performance of their explainer by comparing it against several baselines on two metrics. They also show evidence that the navigator can efficiently guide the search process. Finally, they show a case study, where they demonstrate the correctness of the computed explanations.

After the original paper was published, Chen & Ying (2023) proposed another approach, termed Temporal Motifs Explainer (TempME). TempME is a T-GNN explanation method, which trains a generative model on the input data, coupled with the target model's predictions to generate temporally local and consistent subgraphs of the input graph, called motifs, with respect to some target event. TempME is shown to outperform the T-GNNExplainer on nearly all datasets and metrics, making it a superior approach to the one discussed here. However, we do not consider TempME in this report, as it was published after the original paper and is not within the scope of this work.

**Motivation**   The growing number of applications for T-GNNs outlines the need for specific explanation methodology. Xia et al. (2023) not only provided a framework for the generation of such applications, but also open-sourced the code foundation for the study.

We endeavoured to reproduce their experimental results and expand upon their findings. We encountered several inconsistencies between the provided code base and the authors' claims, which we sought out to interpret and compare with our experimental results, as well as to provide a better foundation for future parties to readily use the proposed framework for model explanations.

**Structure of the Paper**   In this reproducibility study, we aim to replicate the aforementioned claims of Xia et al. (2023). Additionally, we implement and evaluate three different navigator methods and conduct an in-depth analysis of two hyper-parameters, regarding their effect on the running time and correctness of the explanations. In the following sections, we state the scope of our report, describe the methodology proposed by the original authors, as well as our extensions to it. Then, we show our experimental setup and show our results. Finally, we summarise our results, highlighting the strengths and weaknesses of this method. We conclude our work with a short feasibility study and motivate future work.

## 2   Scope of Reproducibility

The main contribution of the original paper (Xia et al., 2023) is an instance-level post-hoc explanation method, which utilizes an explorer-navigator framework to efficiently compute sparse explanations of a single prediction generated by a target temporal graph neural network (T-GNN). More precisely, they propose the **T-GNNExplainer** model, which performs a guided Monte Carlo Tree Search (MCTS) to find probable explanations. The authors further claim that this method is first-of-its-kind and provide an experimental evaluation on two real-world and two synthetic datasets, using two different temporal graph neural networks (TGN by Rossi et al. and TGAT by Xu et al.). The authors also propose several different baselines and compare their performance using a metric called Area Under the Fidelity-Sparsity Curve (AUFSC).

In this report, we aim to reproduce the following claims made in Xia et al. (2023):

1. Experimental results demonstrate that T-GNNExplainer can achieve superior performance with up to 50% improvements in Area under Fidelity-Sparsity Curve, concerning their proposed baselines.

2. The navigator helps to bias the search process, significantly reducing the search time and improving the performance of the explorer.

3. The highly accurate explanations demonstrate the explainer can find an exact influential event set.

In addition to reproducing the aforementioned claims of the authors, we also perform a number of our own experiments to better understand the capabilities and limitations of the T-GNNExplainer method. These extensions are the following:

1. We note a discrepancy between the navigator method described in the paper and the provided implementation. We implement and experiment with both.

2. We propose a new temporal graph explanation method, deriving inspiration from the original authors' proposal for an MCTS explorer with navigator neural network and compare its performance with that of the original T-GNNExplainer model.

3. We experiment with the number of candidate events that the explainer models consider during inference, and analyse the runtime-fidelity trade-off as a function of rollouts of the MCTS algorithm.

## 3 Methodology

### 3.1 Problem Formulation

For the sake of our arguments, we define some notation conventions, derived entirely from Xia et al. (2023). A temporal graph $\mathcal{G} = (\mathcal{N}, \mathcal{S})$ can be regarded as an ordered sequence of events (or interactions), $\mathcal{S} = (e_1, e_2, \dots)$. Each event $e_i \in \mathcal{S}$ is a tuple of the form $(u_i, v_i, t_i, att_i)$, where $u_i$ and $v_i$ are the source and target nodes (set $\mathcal{N}$), $t_i \in \mathbb{R}$ is the timestamp of the event and $att_i \in \mathbb{R}^d$ is an attribute vector of $d$ dimensions. Unlike static graphs, where the nodes and edges are fixed, temporal graphs develop over time.

**Temporal Graph Neural Networks** The authors of the paper use the encoder-decoder framework proposed in Kazemi et al. (2020) to define a temporal graph neural network (T-GNN). The encoder learns the dynamic embedding of the nodes and edges (or events) of the input graph, while the decoder uses these embeddings to predict future events. More precisely, a T-GNN is defined as a second-order function

$$f_\theta : \mathcal{G} \to \mathcal{Z} \to \mathbb{R}$$

where $\mathcal{Z}$ is a $d$-dimensional vector space of node and edge embeddings. By writing $f_\theta(\mathcal{G}^i)[e_k]$ we denote the probability that event $e_k$ occurs at time $t_k$ given the history of the graph up to time $t_k$, denoted as $\mathcal{G}^k$, as predicted by the T-GNN model. In the following, we will refer to such models as *target models*.

**Objective** The aim is to discern the key events influencing the prediction of a given model at a specific time step ($t_k$). This involves identifying a smaller, significant portion of the input graph, $\mathcal{R}^k$, which closely mirrors the model's prediction on the full input graph $\mathcal{G}^k$.

**Performance metric** Because the objective is to find the most influential events in the input graph, we usually do not have access to ground truth explanations. The original authors instead use *fidelity* as the main performance metric. Fidelity is the ratio between the prediction of the target model on the entire graph $\mathcal{G}^i$ and the prediction on the explanatory sub-graph $\mathcal{R}^i$. The higher the fidelity, the more accurate the explanation. Note that the full input graph trivially has the highest fidelity. To avoid this, the authors also use *sparsity* as a second performance metric. Sparsity is the ratio between the number of events in the explanation and the number of events in the input. We can combine these two metrics by plotting the maximum achieved fidelity given a certain sparsity constraint, i.e. the fidelity-sparsity curve. Thus, the Area Under Fidelity-Sparsity Curve (AUFSC) is used as the main performance metric. A higher AUFSC indicates better performance.

### 3.2 Temporal Graph Attention (TGAT)

Albeit not explicitly defined in Xia et al. (2023), the TGNN-Explainer model, as well as one of the baselines relies heavily on the Temporal Graph Attention (TGAT) block, which is employed in the architectures of both TGAT and TGN target models (Xu et al., 2020; Rossi et al., 2020). The TGAT block uses a multi-head,

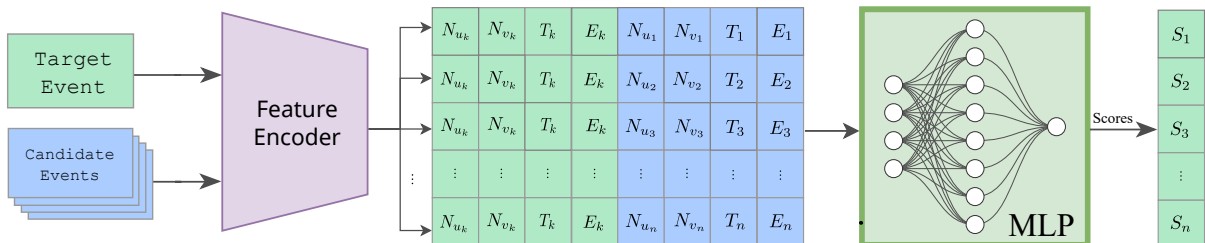

Figure 1: Visualization of the inner structure of the Parameterized Graph Explainer (Luo et al., 2020) adapted for the temporal graph setting. The input representation (source node features, target node features, event time features, and edge features) of the target event is concatenated with the representations for all candidate events considered.

dot-product self-attention layer adapted for a temporal setting. Given a candidate event $e_k$ at time $t_k$, as well as its neighbouring events $e_i \in \mathcal{N}(e_k, t_k) \subseteq \mathcal{S}$, at corresponding times $t_i$. The TGAT block takes as input the hidden representations $[\tilde{h}_k || [\tilde{h}_i]_{e_i \in \mathcal{N}(e_k, t_k)}$ of these events and uses a relative temporal embedding function $\Phi$ to create the input matrix $Z = \left[ [\tilde{h}_k + \Phi(0)] \,||\, [\tilde{h}_i + \Phi(t_k - t_i)]_{e_i \in \mathcal{N}(e_k, t_k)} \right]^T$ (Xu et al., 2020). Then, following the standard definition of the dot-product self-attention mechanism (Vaswani et al., 2017), the TGAT block with $M \geq 1$ heads computes the attention scores $\{\lambda_{e_l}^m\}_{e_l \in N(e_k, t)}$ for head $m \leq M$ as

$$\lambda_{e_l}^m = \frac{\exp(\mathbf{q}^T K_l)}{\sum_{m=0}^n \exp(\mathbf{q}^T K_m)},$$

where $\mathbf{q} = Z_l W_Q$ is the query vector and $K_l$ is the $l^{\text{th}}$ row of the key matrix $K = ZW_K$. Next, we use the value matrix $V = ZW_V$ to obtain the output $\text{Attn}(Q, K, V)_l^m = \lambda_{e_l}^m V$. The weight matrices $W_Q, W_K$ and $W_V$ are learned projection matrices (Vaswani et al., 2017), the superscript $()^m$ refers to the index of the attention head $m \leq M$. Finally, the TGAT block concatenates the outputs of the $M$ different attention heads and feeds them through a feed-forward neural network $g(\cdot, \theta)$ with ReLU activations to obtain the output hidden representation

$$\tilde{h}_k = \text{ReLU}(g(\text{Attn}^{1:M}(Q, K, V)_k, \theta)).$$

Ultimately, the TGAT *model* contains $L > 0$ such TGAT blocks, one after the other. The output of the $L^{\text{th}}$ block is the hidden representation $\tilde{h}_k$ of the target event, the outputs of the $(L-1)^{\text{st}}$ block are the hidden representations of its neighbourhood $e_i \in \mathcal{N}(e_k, t_k)$ and so on. So, a TGAT model with $L$ blocks considers the $(L-1)$-hop neighbourhood of the target event. For further details, please refer to Xu et al. (2020).

**Aggregated Attention Scores** The authors of Xia et al. (2023) use the attention weights $\lambda_{e_i}$ in their implementation. More precisely, given a set of *candidate events* $\mathcal{C}_k$ in the $(L-1)$-hop neighbourhood of a given target event $e_k$, they compute the mean attention scores on these candidate events as

$$\Lambda_{e_i} = \frac{\sum_{l=1}^L \lambda_{e_i}^l}{\sum_{l=1}^L \delta_{e_i}^l}$$

for all events $e_i \in \mathcal{C}_k$. Here, $\lambda_{e_i}^l$ is the attention score of event $e_i$ in the $l^{\text{th}}$ TGAT block, averaged over the attention heads, or 0 if the event is not considered in this block. Similarly, $\delta_{e_i}^l = 1$ if the event is present in the $l^{\text{th}}$ block and $\delta_{e_i}^l = 0$ otherwise. Details on how the candidate events are selected are discussed in Section 3.3.

These $\Lambda_{e_i}$ values form the basis of the proposed **ATTN** baseline and are also adapted for use in their interpretation of the **PGExplainer** (Luo et al., 2020) baseline. Furthermore, the implementation for PG-Explainer is shared by the *PGNavigator* which outputs scores that guide the MCTS expansion process. We elaborate on these in more detail in sections 3.3 and 3.4 respectively.

### 3.3 Model descriptions

In this section, we describe the explorer-navigator framework for temporal graphs, which is the main theoretical contribution of the paper. We also describe the T-GNNExplainer model and the various baselines they proposed.

**Explorer-Navigator Framework** In the previous section, we defined explanations as sub-graphs of the input graph, which maximizes the mutual information between the prediction of the target model on the sub-graph and the prediction on the entire graph. However, this is a combinatorial optimization problem, which is intractable to solve in practice. Instead, the authors use an approximate solution, based on the explorer-navigator framework. The explorer is a Monte Carlo Tree Search (MCTS) algorithm guided by a pre-trained navigator.

The navigator is a feed-forward neural network, which assigns importance scores to event pairs of the input graph. More precisely, given a sequence of events $\mathcal{S}$, the navigator is defined as a function:

$$g : (\mathcal{S} \times \mathcal{S}) \to \mathbb{R}.$$

The T-GNNExplainer, given a set of candidate events, spatially and temporarily close to the target event, uses this navigator to compute pair-wise importance scores for each candidate and the target event.

Finally, the explorer is a search algorithm, which considers different subsets of the candidate events to come up with viable explanations. In (Xia et al., 2023), the authors use the Monte Carlo Tree Search (MCTS) for this. The MCTS algorithm is used to find the best path in a tree, where each node represents a state of the system and each edge represents a possible action. In our problem, a state (or tree node) is a collection of events, and each edge means the removal of a certain event from this collection. Thus, a tree node of $n$ events has $n$ distinct children. The MCTS algorithm finds the (approximate) best path in this tree with the highest associated score. The score function is simply the fidelity of a set of events in a given node. More precisely, given the input graph up to time step $t$ ($\mathcal{G}^t = (V, \mathcal{S}^t)$), target model $f(\cdot)$, and a set of events present in some tree node $\mathcal{N}^i$, ($\mathcal{R}^{t,i} \subseteq \mathcal{S}^t$), the fidelity of this set of events is the binary cross-entropy between the target's prediction on the entire input against the prediction on the given subset:

$$\text{fid}(\mathcal{G}^t, \mathcal{R}^{t,i}) = \begin{cases} f(\mathcal{R}^{t,i})[e_t] - f(\mathcal{G}^t)[e_t] & \text{if } Y_t = 1 \\ f(\mathcal{G}^t)[e_t] - f(\mathcal{R}^{t,i})[e_t] & \text{if } Y_t = 0 \end{cases},$$

where $Y_t$ is the ground truth label whether event $e_t$ occurs or not. This algorithm is deterministic because the fidelity function is deterministic given a set of events and a fixed target model $f(\cdot)$. Always selecting the path with the highest score is called *exploitation* and traversing different paths is called *exploration*. For the search algorithm to succeed, these two strategies should be balanced. The Upper Confidence bound applied to Trees (UCT) (Kocsis & Szepesvári, 2006) is a commonly used method to address this problem. Given a tree node $\mathcal{N}^i$, the UCT algorithm selects the best action, i.e. the removal of some event $e^*$, from the set of events available in $\mathcal{N}^i$, by solving:

$$e^* = \underset{e_j \in \mathcal{C}(\mathcal{N}^i)}{\arg\min} \left( \frac{c(\mathcal{N}^i, e_j)}{n(\mathcal{N}^i, e_j)} + \lambda \frac{\sqrt{\sum_{e_l \in \mathcal{C}(\mathcal{N}^i)} n(\mathcal{N}^i, e_l)}}{1 + n(\mathcal{N}^i, e_j)} \right).$$

Here, $\mathcal{C}(\mathcal{N}^i)$ is the set of nodes already expanded in node $\mathcal{N}^i$, $n(\mathcal{N}^i, e_j)$ is the number of times event $e_j$ was previously removed from node $\mathcal{N}^i$ and $c(\mathcal{N}^i, e_j)$ is the cumulative reward of removing $e_j$. The first term selects the event with the highest cumulative reward (*exploitation*), while the second term selects the event with the highest uncertainty (*exploration*). The parameter $\lambda$ controls the trade-off between the two. We refer to a single path selection procedure, from the root to a leaf, as *rollout*. During search, a predefined number of such rollouts are executed. In each subsequent rollout, the UCT algorithm considers all the previous iterations to find the best path. Ultimately, we select the highest fidelity tree node as the explanation.

**T-GNNExplainer** The T-GNNExplainer model utilizes this explorer-navigator framework to compute explanations for a given prediction made by a target model. When called with a target event, it selects a

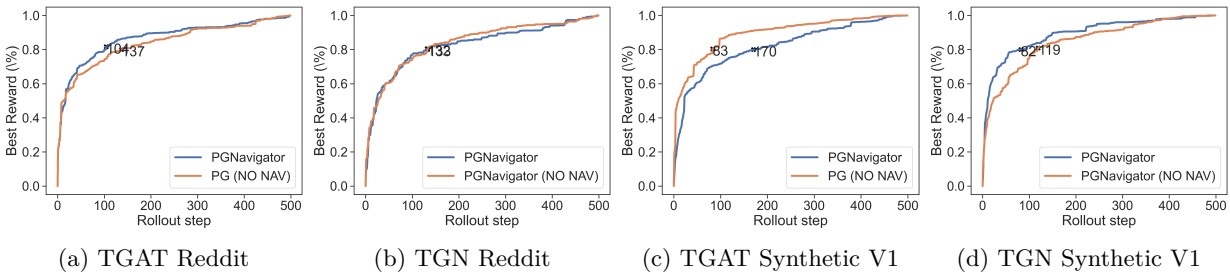

Figure 2: Best fidelity-rollout curves with and without the navigator on TGAT (a, c) and TGN (b, d) models evaluated on Reddit and Synthetic V1 datasets. We denote the points where the fidelity first reaches 80% best fidelity. We can see that the navigator helps the search process in the Reddit dataset, but hinders the Synthetic V1 dataset, especially on the TGN backbone.

pre-defined number of candidate events (sometimes this number is referred to as *threshold*) and computes the importance scores using the navigator. Finally, the MCTS algorithm is used to compute an explanation. It considers the candidate events in decreasing order of importance.

## 3.4 Baselines

In addition to the T-GNNExplainer model, a first-of-its-kind temporal graph explainer, the authors propose several baseline TGNN explanation methods to compare their performance. A crucial difference between the baselines and the T-GNNExplainer is that the latter is search-based, while the former are not. The baselines are as follows:

**PBONE**  Obtain explanations by local perturbations of the candidate events. This is based on the assumption that the more influential an event is, the more sensitive the target model will be to small changes in that event.

**ATTN**  The computed mean attention scores of the target model, $\{\Lambda_{e_i}\}_{e_i \in C_k}$, as described in Section 3.2. This method is based on the assumption, that the more influential an event is, the more attention the target model pays to that event.

**PGExplainer**  is a modified version of the Parameterized Graph Explainer model, first proposed in Luo et al. (2020). At the core of it is a Multi-Layer Perceptron (MLP), denoted as $g(\cdot, \cdot, \theta)$, whose parameters, $\theta$, are trained to predict pair-wise importance scores of input events. The authors deviate from Luo et al. (2020) in two crucial details. First, they adapt this model to the temporal setting, via harmonic temporal embeddings. Secondly, they use the predictions of the MLP as soft-masks over the input graph, and then similarly compute the mean attention scores as in the ATTN baseline. Formally, given a target event $e_k$ and corresponding set of candidate events $\mathcal{C}_k$, for each candidate $e_i \in \mathcal{C}_k$, the importance score is computed as

$$s_{e_i} = \Lambda_{e_i}^{SM},$$

where $\Lambda_{e_i}^{SM}$ is the mean attention score (as defined in Section 3.2) of the target model on event $e_i$, when evaluated on the *soft-masked* input $\{h_{e_i} \cdot g(h_k, h_i, \theta)\}_{e_i \in \mathcal{C}_k}$.

### 3.4.1 Navigators

While the previous sections give an intuitive understanding of the explorer-navigator framework and the proposed baselines, our main contribution to this project is the comparison of three different navigator methods. In this section, we aim to introduce each and clarify the key differences between them.

**MLPNavigator**  The authors of the paper Xia et al. (2023) propose a navigator model, essentially a feed-forward neural network, computing pair-wise importance scores between the input representations of the

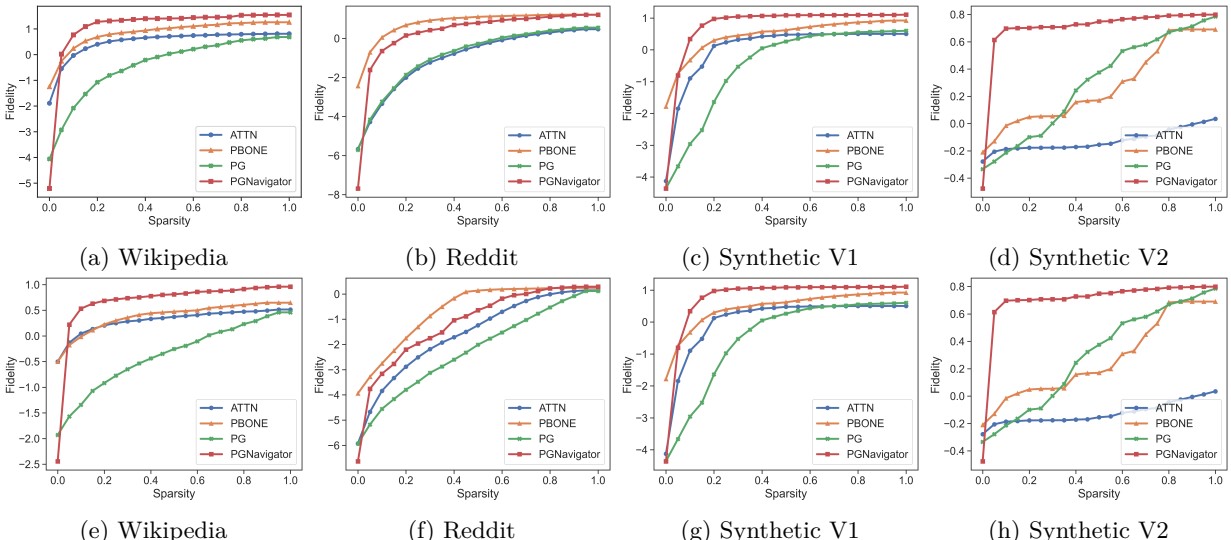

Figure 3: Fidelity-sparsity curves for TGAT (a-d) and TGN (e-h), threshold 20. By comparing the area under these curves, one can evaluate the performance of different explainer models. The greater the area, the better the method at finding high-fidelity explanations across different sparsity constraints. We can see that the proposed T-GNNExplainer (red) out-performs the baselines in almost all cases.

target event and the candidate events. Formally, it is a function $g(\cdot, \cdot, \theta)$, where $\theta$ are the learned parameters of the network, equivalent to the pre-trained parameters of the PGExplainer baseline. Given a target event $e_k$ and a set of candidate events $e_i \in C_k$, the MLPNavigator computes

$$s_{e_i} = g(h_k, h_i, \theta),$$

where $h_k, h_i$ are the input representations. We have additionally provided a visualization for how this navigator works in practice in the TGNN setting in Figure 1. In the following, we will refer to this method as MLPNavigator. Note that, albeit this definition is aligned with the definition given in Xia et al. (2023), in practice, they used a slightly different method, which we discuss next.

**PGNavigator** As it became evident from the provided code base and subsequent reproduction results, the navigator method is not exactly as described by Xia et al. (2023). In fact, the candidate importance scores are obtained precisely using the method of the above-defined version of the PGExplainer baseline. To disambiguate this approach from the MLPNavigator, in our work, we refer to this navigator as PGNavigator. See Xia et al. (2023, sec. 4.2) for more details.

**DotProductNavigator** Finally, to further evaluate the performance of the navigator, we propose a different approach, which we call **DotProductNavigator**. In this method, instead of computing an inductively learned pair-wise importance score between *input event pairs*, we compute the pair-wise dot product of the *output embedding* of the target model's *encoder* module. As such, given a target event $e_k$ and candidate events $e_i \in \mathcal{C}_k$, we compute the importance score as

$$s_{e_i} = \tilde{h}_{e_k}^T \tilde{h}_{e_i},$$

where $\tilde{h}_{e_k}, \tilde{h}_{e_i}$ are output hidden representations of the target model's encoder module (see Section 3.2. We hypothesize that a high performance on this method shows that the target model learns to encode the input graph in a way, such that excitatory events appear close to the target event in the embedding space. This is a desirable property since it provides an intuitive understanding of the model and its training goals, even beyond the scope of local interpretability.

## 4 Experimental setup and code

We based our experiments on the code provided by the authors[1]. This repository contains the code for the target models, the T-GNNExplainer model and the baselines. Shell scripts and configuration files are also supplied, to help run the experiments, train the target models and pre-process the datasets.

In the following, we describe the datasets we used and the results we obtained. We also discuss the computational requirements and list the hyper-parameters we used in Appendix A.1 and A.2 respectively.

In addition, we provide detailed documentation on running the explainer pipeline, structured executable scripts to reproduce all findings, as well as several optimizations. Full details are available in Appendix A.3.

### 4.1 Datasets

The authors of the paper evaluated their explainer model on two real-world and two synthetic datasets. The real-world datasets are the following:

**Reddit** [2] consists of posts made by the 10 000 most active users over a one month period on the 1000 most active subreddits; contains a total of 672 447 interactions between users and subreddits(Kumar et al., 2019).

**Wikipedia** [3] is a collection of edits on the 1 000 most edited wikipedia pages over a one month period. It contains 8 227 editors and 157 474 interactions in total (Kumar et al., 2019).

**Synthetic V1 and V2** The synthetic datasets are both generated by the multivariate Hawkes process (Hawkes, 1971) with different hyper-parameters. Hawkes processes are point processes, which model the occurrence of events over time. The events are generated by simple event-relation graphs, where edges roughly correspond to probabilities that the two nodes interact at a given time step. The authors propose two different event-relation graphs and generate two datasets of approximately 10 000 timestamps. They make use of the *tick* library (Bacry et al., 2018) to model the Hawkes processes.

The synthetic datasets fit well within the scope of this project, as they inherently contain ground truth data due to the generating hyperparameters. The authors make use of the datasets particularly within the case study which aims to evaluate the accuracy of the proposed explanations.

## 5 Results

### 5.1 Results reproducing the original paper

We list the main results reproducing the authors' described experiments in Table 1a and 1b. For a detailed comparison, we also provide the authors' original results as well as the relative divergence ratio of our results and theirs in Tables 4 and 5. Furthermore, we reproduce the fidelity-sparsity curves for the different datasets and the case study of ground truth explanations, as provided by the original authors.

Generally, we observe a large variance in the results of the proposed evaluation metrics. The Best Fidelity in particular seems to be particularly prone to large differences from setting to setting. The AUFSC metric seems less volatile, but the observed variance remains high, which complicates an accurate evaluation of the proposed method against the baselines. In the following, we attempt to summarise our results concerning the three main claims we laid out in Section 2.

**Claim 1: Superior performance compared to baselines** We note approximate relative difference between the reproduced and original results of the proposed explainer (PGNavigator) of $-19\%$, and $-46\%$ on the TGAT, and the TGN target models, respectively. Regardless, the explainer still outperforms the other baselines with gains across the Wikipedia, Reddit, Synthetic V1 and V2 datasets on AUFSC (Best

---

[1] https://openreview.net/forum?id=BR_ZhvcYbGJ
[2] http://snap.stanford.edu/jodie/reddit.csv
[3] http://snap.stanford.edu/jodie/wikipedia.csv

|  | Wikipedia | | Reddit | | Synthetic V1 | | Synthetic V2 | |
| --- | --- | --- | --- | --- | --- | --- | --- | --- |
|  | Best FID | AUFSC | Best FID | AUFSC | Best FID | AUFSC | Best FID | AUFSC |
| ATTN | 0.622 | 0.179 | 0.060 | -0.085 | 0.812 | 0.514 | 0.466 | -0.936 |
| PBONE | 1.024 | 0.628 | 0.708 | 0.442 | 1.260 | 0.869 | 1.212 | **0.817** |
| PG | 0.678 | -0.306 | 0.746 | 0.228 | 0.685 | -0.375 | 0.548 | -0.829 |
| PGNavigator | 1.155 | **0.842** | 0.789 | **0.720** | **1.513** | **1.143** | 1.155 | 0.444 |
| MLPNavigator | **1.182** | 0.777 | **0.795** | 0.605 | 1.395 | 0.881 | 1.162 | 0.368 |
| DotProductNavigator | 0.987 | 0.469 | 0.783 | 0.713 | 1.253 | 0.598 | **1.223** | 0.596 |

(a) † Best Fidelity (↑) and AUFSC (↑) achieved by each explainer on TGAT model.

|  | Wikipedia | | Reddit | | Synthetic V1 | | Synthetic V2 | |
| --- | --- | --- | --- | --- | --- | --- | --- | --- |
|  | Best FID | AUFSC | Best FID | AUFSC | Best FID | AUFSC | Best FID | AUFSC |
| ATTN | 1.502 | 0.889 | 1.660 | -0.677 | 0.515 | 0.313 | 0.145 | -1.546 |
| PBONE | 1.814 | 0.886 | 2.813 | -0.290 | 0.648 | 0.393 | 0.244 | **-0.629** |
| PG | 1.436 | 0.092 | 1.112 | -1.903 | 0.460 | -0.367 | 0.117 | -2.209 |
| PGNavigator | 1.824 | 1.445 | 2.816 | **1.916** | **0.960** | **0.700** | 0.289 | -1.120 |
| MLPNavigator | **2.037** | **1.611** | 2.443 | 0.898 | 0.829 | 0.444 | 0.239 | -1.492 |
| DotProductNavigator | 1.219 | 0.053 | **2.975** | 0.404 | 0.943 | 0.317 | **0.291** | -1.391 |

(b) † Best Fidelity (↑) and AUFSC (↑) achieved by each explainer on TGN model.

Table 1: Results for each target setting. Best score is emphasized and second-best is underlined. We can see that the T-GNNExplainer variants (last three rows) dominate across almost all datasets and target models.

Fidelity) of 34%(13%), 62%(5%), 32%(20%), and -5%(-84%) on the TGAT model, as well as 63%(1%), 321%(1%), 78%(48%), -78%(14%) on the TGN model. We further note that the second-best baseline in the original paper, PGExplainer does not perform nearly as well as the authors report. The performance of the perturbation baseline PBONE, within our experiments, diverges from the original paper, where it performed marginally worse than all other explainers. The ATTN baseline performs identically in most settings.

Performance within the constraints of fidelity-sparsity (Figure 3) appears comparable to the original study. Notably again, PBONE, which in the original paper was the worst performer across the various settings, performs competitively with the other baselines and marginally surpasses them on AUFSC.

**Claim 2: Navigator improves performance**   The authors originally claim that the navigator improves the performance of the explorer and provide quantitative evidence that the explorer, guided by the navigator crosses a certain fidelity threshold much earlier than the explorer without the navigator on all datasets and target models. They show the measured running time of the explorer to find an explanation, whose fidelity is at least 80% of the best fidelity found overall. Here we note that the running time in seconds is inherently volatile and largely depends on the hardware used. Instead, we plot the achieved highest fidelity explanation over the number of rollouts of the MCTS algorithm.

Figure 2 shows representative examples of our results. We observe that in the case of real-world datasets (Wikipedia and Reddit), the navigator improves the performance. However, this is not the case for synthetic datasets, where using the navigator causes the explainer to reach the same fidelity more slowly. This is an interesting result and may be explained by noting that the synthetic datasets are represented on only 4 dimensions in the input, whereas the real-world datasets have much higher embedding dimensionality. This can cause the navigator to overfit the training data and thus provide less useful guidance to the explorer. This result is contrary to what was reported in Xia et al. (2023), even though we used the same hyper-parameters to initialize it (see Appendix B).

**Claim 3: Sparse and accurate explanations**   We show explanations produced by the authors' proposed explainer model in Figure 4. We replicated the conditions presented in (Xia et al., 2023,   sec. 5.5). Namely,

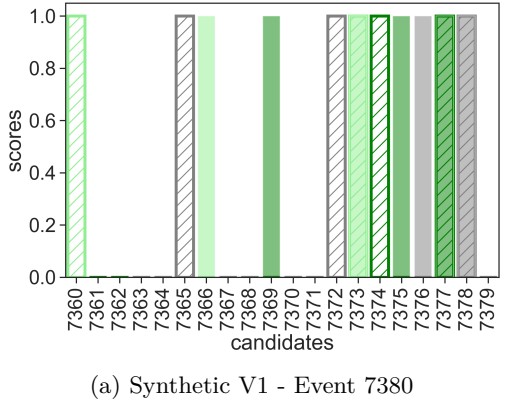

(a) Synthetic V1 - Event 7380

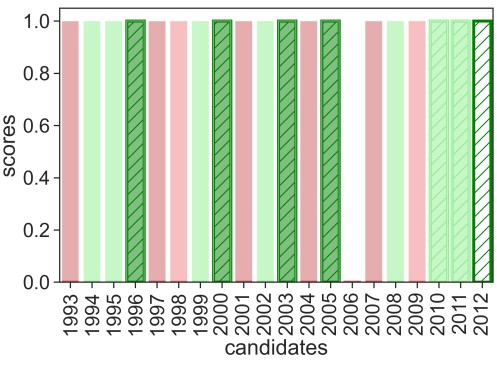

(b) Synthetic V2 - Event 2013

Figure 4: Explanations produced by the authors' proposed explainer model (using PGNavigator). The bar plot shows the candidate events the explainer model considered and sets all events that are part of the highest fidelity explanation to 1 and all others to 0. Patterned bars represent authors' claimed results whilst bars with background fill color refer to our results. Dark- and light-green columns represent excitatory events, while grey and red columns represent unimportant and inhibitory events, respectively. Both explanations were produced for the TGAT model. Note that while on the left, the reproduced explanation is similarly sparse and selects mostly excitatory events, this is not the case for our reproduction on the right, where all except two events were selected, many of which are inhibitory.

we evaluate the explainer over the TGAT model, on event number 7380 of the Synthetic V1 dataset and event number 2013 of the Synthetic V2 dataset. We can see that the explainer model can produce sparse and accurate explanations in the first case, but fails to do so in the second case. This is inconsistent with the author's results, where the explanations were in line with their claims in both cases.

## 5.2 Results beyond the original paper

As mentioned in Section 2, on top of the reproduction of the original paper, we also conduct a number of additional experiments. We motivate and describe these experiments and provide the results we obtained.

### 5.2.1 Navigators

Tables 1a and 1b summarise the performance of the T-GNNExplainer, executed using different navigator implementations, as well as the performance of the baselines. We evaluated them on two metrics, namely the highest fidelity explanation computed and the area under the fidelity-sparsity curve. In general, we can see that the T-GNNExplainer variations outperformed the baselines on both metrics.

Further, we observe that the navigator proposed by the original authors (PGNavigator) marginally outperforms the other two in most cases. The navigator described by the authors (Xia et al., 2023) (MLPNavigator) is competitive.

Finally, the DotProductNavigator performed similarly competitively, albeit with a very weak performance for the Wikipedia with TGN backbone setting. This suggests that the candidate events do in fact appear similar to the target event in the learned embedding space of both target models for many cases and merits further exploration as an explainer method. Overall, we can conclude, as per the authors' original claim, the PGExplainer model[4] can be used as a navigator for an MCTS algorithm to efficiently guide the search process for accurate and sparse explanations in the given settings.

---

[4]Note the identical inference method as PGNavigator.

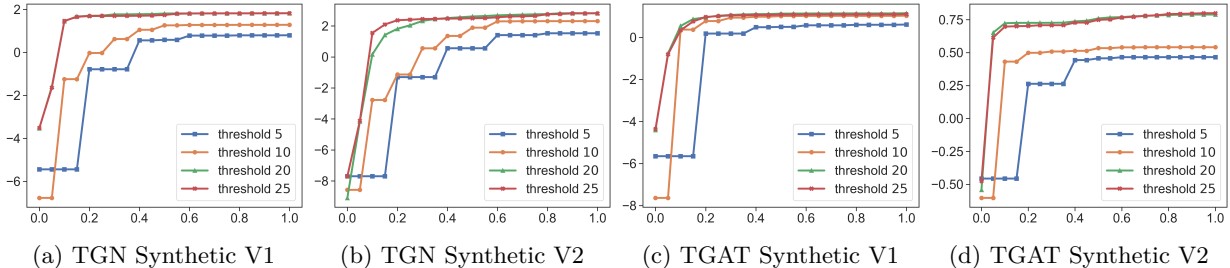

Figure 5: Fidelity-sparsity curves for the two target models on the synthetic datasets, at considered candidate events threshold $(5, 10, 20, 25)$. The performance of the explainer grows proportionally with the number of candidate events until 20 but the performance at 25 candidate events is only marginally better than at 20.

### 5.2.2 Number of Candidates and Rollouts

As mentioned in Sections 2, we conducted hyper-parameter tuning experiments on the number of candidate events the explainer considers. This is a significant hyper-parameter, as it controls the effective search space the explainer has access to for finding probable explanations. Intuitively, a higher number of candidate events should lead to a more accurate explanation but also increase the time complexity of the model. The authors of Xia et al. (2023) claim to have used the most recent $\leq 25$ events in the $k$-hop neighbourhood of the target event, but we also performed experiments with $5, 10, 15$ and $20$ such events. We show representative example the resulting fidelity-sparsity curves in Figure 5.

In our experiments, we found that the AUFSC score grows proportionally to the number of candidate events, up to 20, but at larger numbers this growth quickly becomes insignificant. To explain this result, we note that both synthetic datasets contain only four nodes and their event-relation graphs are almost trivial. As a result, the current event is always largely decided by the short-term history of the event sequence preceding the target. We also note that the optimal value for this parameter is likely much higher for real-world datasets, which are much more intricate and complex temporal graphs. We hypothesize, that the optimal value for this parameter is closely related to the complexity of the underlying dataset.

Finally, we wanted to find an appropriate number of rollouts. This number directly affects the length of the search process and subsequently limits the proportion of the search space the MCTS algorithm explores. In our analysis, we wanted to quantify this tradeoff. Figure 2 shows the highest fidelity reached by the explorer over 500 rollouts. We find that these curves roughly follow a cumulative Pareto distribution and 80% of the best fidelity is reached within the first $\approx 200$ rollouts.

## 6 Discussion

Overall, we conclude that the results of the paper are not completely reproducible, partly due to the incomplete information about the random number generator used in their experiments. Regardless, our obtained values are marginally within the original authors' reported results.

Concerning the authors' original claims, we found that

1. The T-GNNExplainer outperforms the other baselines across nearly all datasets and metrics. However, the performance of the proposed explainer model is not as high as the authors claim. We partly attribute this finding to the high volatility of the proposed metrics.

2. The navigator increases the performance of the explorer, but only for real-world datasets. For synthetic datasets, the navigator hinders the search process. We surmise that the navigator could possibly be overfitting to the training data.

3. The T-GNNExplainer can produce sparse and accurate explanations, but not in all cases. We partly attribute this to differently trained target models (See Appendix 3) and note the need for further exploration.

Additionally, we evaluated three different navigator implementations and found that (i) the provided implementation is marginally better than what was described in the paper and (ii) using dot product similarity between candidate and target events performs particularly badly on the AUFSC metric. This suggests that the importance of candidate events concerning a target event does not correlate well with their similarity in the embedding space of the target models.

Finally, we also experimented with the number of candidate events the explainer has access to and found that in the case of the synthetic datasets, considering the 20 most recent events as candidates yields the best runtime-fidelity tradeoff. Furthermore, we also found that $> 80\%$ of the best fidelity is reached within the first 200 rollouts of the MCTS algorithm.

**What was easy**  The paper provided an in-depth explanation of their methodology in a precise and concise manner. This made it easier to understand the theoretical background of their approach. Furthermore, the authors provided a code base, which saved a lot of time in our experiments.

**What was difficult**  Understanding the provided code base proved to be difficult, as it not only did not follow the conventions introduced in the paper, but it also was not entirely consistent with the descriptions of the paper, which made it difficult at times to reproduce their results. Part of our contribution is that we ensure the code base can be set up and executed without errors. Moreover, we also provide some running time optimizations to the MCTS algorithm and disambiguate the naming conventions. We provide more details in Appendix A.3.

## 6.1  Feasibility

In this section, we attempt to summarise our findings such that the reader can evaluate whether the T-GNNExplainer model is feasible for real-world applications.

Earlier, in Section 3.4.1 we noted that the PGNavigator is equivalent to the PGExplainer baseline. In our experiments discussed in Section 5.1, we found that this search process improves the fidelity of the explanations by  30% on and also brings a  30% increase on the AUSFC metric, on real-world datasets. On the other hand, we also noted in Section 5.2, that executing the MCTS algorithm takes significant time and this time grows exponentially in the number of candidate events considered. Conversely, the inference time of the PGExplainer model only grows linearly in the number of candidates.

Further, it is important to reiterate, that the PGNavigator, implemented in the supplementary material of Xia et al. (2023) is not equivalent to PGExplainer, originally described in Luo et al. (2020). The most crucial difference is that this version derives its output from the aggregated attention scores $\Lambda_{e_i}$, as we discussed in Sections 3.2 and 3.4.1. Here we note that this limits the applicability of the T-GNNExplainer model to the set of models that rely on architectures similar to that of the TGAT architecture (Xu et al., 2020).

Overall, we recognise the significant improvement in the fidelity of the explanations computed by the T-GNNExplainer, but also note a significant running time overhead compared to less accurate, inference-only solutions. Thus, in a large-scale deployment, it may be advisable to use some ensemble of inference-only explainers and only invoke the T-GNNExplainer for a select few cases, where high fidelity is instrumental.

Finally, we note that the TempME model (Chen & Ying, 2023) is a more recent approach, which has been shown to outperform the T-GNNExplainer model in nearly all datasets and metrics, producing more interpretable explanations.

## 6.2  Future work

Earlier, we noted that for synthetic datasets, observing the 20 most recent events yields the highest runtime-fidelity trade-off. We also note that this may not be the case for real-world datasets. We hypothesize that this parameter is an inherent characteristic of the underlying dataset and the optimal value of this parameter may be orders of magnitude larger for more complex temporal graphs. Quantifying the number of candidate events needed for good explanations as a function of the input graph's complexity would not only help with explanations but also introduce a novel aspect of characterising temporal graphs and T-GNNs.

## 6.3 Communication with original authors

We reached out to the authors for a more detailed description of their hyper-parameters and to inquire about follow-up research conducted since the publishing of the original article. We have not received any replies.

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

# A  Appendix

## A.1  Computational requirements

To run our experiments, we use the following hardware. Reported results will be indicated with one of the associated symbols when shown in tables.

**MBP**† Intel Core i9-9900H CPU @ 2.30GHz and 16GB RAM, AMD Radeon Pro 560X GPU

**Snellius**‡ Intel Xeon Platinum 8360Y (2x) 36 Cores/Socket 2.4 GHz, 32GB RAM, NVIDIA A100

**Training the target models**  Training the target models requires significant resources and time. For this reason, we decided to only train them once. Additionally, we release the learned parameters of these models so that they can be used for further experiments.

## A.2  Hyper-parameters

Finally, we list the hyper-parameters we used in our experiments with the T-GNNExplainer model. Hyper-parameters related to training the target models are listed in Appendix B.

**Navigator**  The navigator is a two-layer perceptron network with ReLU activations and 128 hidden units. Timestamps are encoded using a two-layer attention architecture and a harmonic encoder, as described in both Rossi et al. (2020) and Xu et al. (2020). The datasets are randomly split into training, validation and test which consist of 70%, 15% and 15% sets of the data respectively. The learned model weights are shared across the authors' and our navigator implementations, as well as with the PGExplainer baseline.

**Explainer**  We evaluate the T-GNNExplainer on three different navigators (see Section 3.4.1). We set the number of rollouts of the MCTS algorithm to 500, and the $\lambda$ parameter of the UCT algorithm, controlling exploration-exploitation trade-off, to 5. These hyper-parameters are the same as the ones used by the authors. Additionally, to reduce the cost of our experiments, we randomly selected only 100 target events for explanations, instead of 500 as used by the authors. This is a minor divergence since results are aggregated over all explained events. Finally, we noticed evidence in the provided code base and in the paper, that the authors evaluated all explainer models (both PGNavigator and the baselines) with the 20 most recent candidate events. This is a divergence from the paper, where 25 is claimed for this parameter. We experiment with both values and additionally explore the effects of considering only 5 and 10 candidate events. We provide our analysis on the effect of this parameter in Section 5.2 and in Appendix A.6.

**Random seed**  Lastly, it is important to note that although the pre-training of the navigator and the target models involves randomly initialized parameters, at inference time, both the T-GNNExplainer and the baselines are deterministic. Moreover, both the training of the target models and executing a complete experiment took significant time and resources. For these reasons, we did not conduct any further experiments with differently initialized random number generators.

## A.3  Changelog

In this section, we highlight the most important changes we have performed on the code base. It includes fixes to some of the discrepancies between the paper and the provided code base, as well as some minor optimizations and non-functional changes to the code base.

**Number of candidate events**  As we noted in Section 3.3, according to the paper, both the baseline explainers and the T-GNNExplainer model consider the 25 most recent events in the $k$-hop neighbourhood of the target event. However, in the provided code base, this parameter is hard-coded to 20. The value of this parameter is significant, as it controls the effective size of the search-space that the explainers condider. In our work, however, we could not initially determine whether the reported experimentation results were

produced using the value of 20 or 25. For this reason, we experimented with both. Additional information is provided in Section 5.2.

$\lambda$ **parameter**    In the article, the authors claim that they used $\lambda = 5$ for the UCT algorithm. However, in the provided code base, it is set to 100. For the interested reader, this parameter is called `c_punct` in the provided code base. We set this parameter to 5 for all our experiments.

**Further changes**    On top of the above mentioned inconsistencies, we also performed small optimizations and some improvements to the code base. These include:

- In the original code base there was no clear distinction between the PGExplainerTG class and the MLP used as navigator in the authors' explainer model. In fact, the MLP used as a navigator was instantiated by a static method within the PGExplainer class and the pre-trained model weights would be loaded during this process. We slightly modified the code base by introducing a new module `navigators`[5], which now contains three classes: **PGNavigator**, **MLPNavigator** and **DotProductNavigator**. One can switch between using these different navigators by setting the *navigator_type* parameter in the relevant configuration file[6].

- We optimized the MCTS algorithm by removing superfluous linear searches in the node selection algorithm. This optimization lead to a meaningful performance improvement of the TGNN-Explainer of around 30%.

- We introduced a new hyper-parameter, controlling the number of most recent events that the explainers consider. This parameter was hard-coded to 20 in the provided code base.

- The implementation of the PGExplainer in the provided code base was inefficient, in that each time it was called, it would reload the trained model weights from a checkpoint. In the absence of such checkpoint, it would first train the model, and then return from the function. In case of 25 candidate events, this would mean that the PGExplainer would be trained once for the first event and its model weights would be reloaded 24 times. We fixed this by moving this logic to the constructor on the PGExplainer model, where all the necessary information is available.

- Finally, we also noted that in the absence of a pre-trained PGExplainer model, the navigator would not be properly instantiated and the T-GNNExplainer model would fail to run. One possible reason for this bug is that both the architecture and the training procedure of the navigator coincide with the PGExplainer model, and instead of re-implementing the navigator, the authors simply exposed the fully connected layers of the PGExplainer when instantiating the T-GNNExplainer.

### A.4    Runtime Comparison

We list the required wall-time for the explanation of a single event within the framework of the proposed by the authors' baselines, as well as the search-based explorer-navigator framework methods (Table 2).

Notably, the search-based algorithms take considerably longer to be completed due to the tree traversal required, which 1) cannot be reliably parallelized; and 2) requires the computing of the model probabilities along almost all steps of the traversal.

The variant of PGNavigator explainer algorithm that does not use a navigator is only faster in the case of the Synthetic V1 dataset.

### A.5    Performance of the Proposed Navigators

We furthermore provide additional figures in Figure 6 that aim to showcase the performance of the additional navigators we have implemented, as they have been shown to perform close to or even surpass the

---

[5]`tgnnexplainer.xgraph.method.navigators`
[6]`benchmarks/xgraph/config/explainers/subgraphx_tg.yaml`

|  | Synthetic V1 | Synthetic V2 | Wikipedia | Reddit |
|---|---|---|---|---|
| ATTN | 0.009±0.0 | 0.009±0.0 | 0.016±0.0 | 0.034±0.0 |
| PBONE | 0.132±0.0 | 0.132±0.0 | 0.265±0.0 | 0.282±0.0 |
| PG | 0.001±0.0 | 0.001±0.0 | 0.001±0.0 | 0.001±0.0 |
| PGNavigator | 39.956±10.4 | 44.591±13.1 | 43.178±2.9 | 52.496±6.5 |
| PGNavigator (NO NAV) | 33.784±5.9 | 34.971±14.3 | 47.286±12.0 | 80.111±17.8 |
| MLPNavigator | 33.577±7.0 | 43.377±12.3 | 63.063±11.9 | 56.848±7.3 |
| DotProductNavigator | 36.433±7.9 | 40.787±18.4 | 49.003±11.6 | 52.723±6.5 |

(a) † Runtime comparison for a single event explanation with TGAT base model (in seconds).

|  | Synthetic V1 | Synthetic V2 | Wikipedia | Reddit |
|---|---|---|---|---|
| ATTN | 0.006±0.0 | 0.006±0.0 | 0.014±0.0 | 0.029±0.0 |
| PBONE | 0.079±0.0 | 0.079±0.0 | 0.222±0.0 | 0.251±0.0 |
| PG | 0.001±0.0 | 0.001±0.0 | 0.001±0.0 | 0.001±0.0 |
| PGNavigator | 29.252±6.8 | 13.301±1.4 | 45.987±11.5 | 56.312±7.5 |
| PGNavigator (NO NAV) | 17.274±3.8 | 13.239±1.8 | 51.807±11.8 | 56.552±7.9 |
| MLPNavigator | 23.547±8.9 | 14.031±1.6 | 42.573±6.8 | 74.467±18.2 |
| DotProductNavigator | 25.381±4.6 | 12.470±1.5 | 69.283±28.6 | 56.260±8.1 |

(b) † Runtime comparison for a single event explanation with TGN base model (in seconds).

Table 2: We record the execution times of an event-level explanation over 100 different explained events and all proposed evaluation settings. The results are aggregated and an estimate for the required time of a full event explanation is provided. The baseline methods are not search-based, thus the time required is much shorter. Conversely, as the T-GNNExplainer variants are inherently search-based, they perform considerably slower.

implementation of the original authors, as presented in Tables 1a and 1b. The reported results are, as before, with a threshold of 20 candidate events.

The differences between navigators are often not significant. The **MLPNavigator** performs more or less similarly with the original **PGNavigator** proposed by the authors.

The **DotProductNavigator** we propose as an extension to the project also performs similarly and occasionally surpasses the other navigators, but seems to struggle at lower sparsity constraints. Thus, if for performance reasons, a lower rollout number is selected (e.g. 200 instead of 500) for faster performance, relatively lesser final fidelity might be expected, but could potentially perform faster.

However, the relatively high performance of the **DotProductNavigator** still merits further research. In future work within temporal graph explanation, we can disregard the MCTS algorithm, which, by definition attempts to find a sparse subset of events that maximizes fidelity through the use of navigators, and only consider an explainer based on the dot product of target event and candidate events.

## A.6   Results with Threshold 25

We show figures of our experiment using the 25 most recent events as candidates, as the authors claimed in the article in Figure 7. We note, however that the higher number of candidate events considered at each step of the tree expansion leads to an explosion of the number of total candidates considered, which is the primary drawback to this method.

Otherwise, as already noted in Section 5.2.2 and Figure 5, the performance improvement is only marginal.

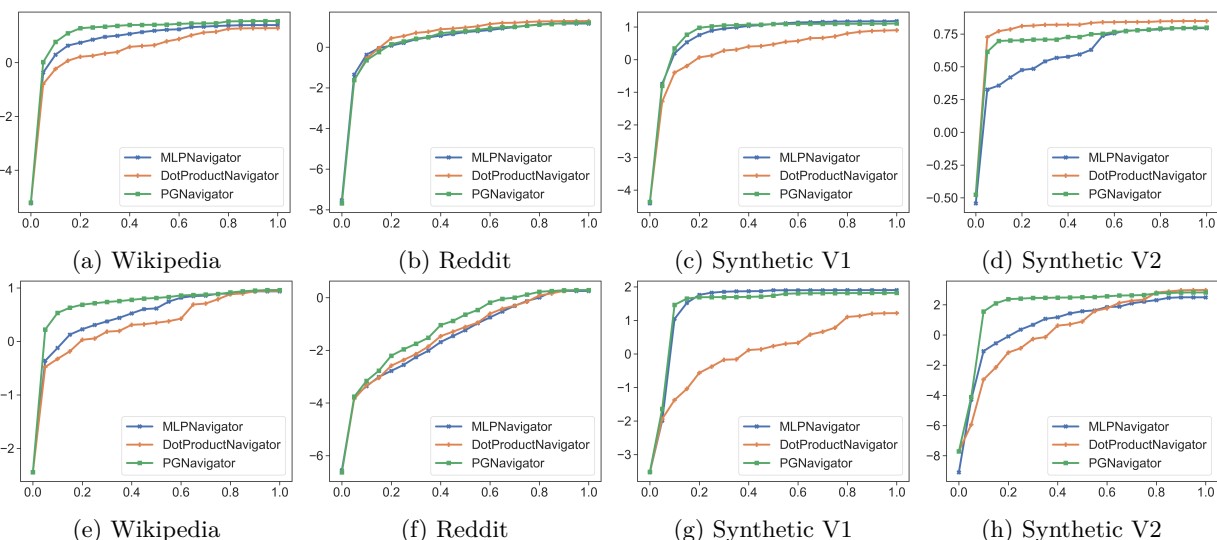

Figure 6: † Results for the different navigators with TGAT (a-d) and TGN (e-h) with threshold 20.

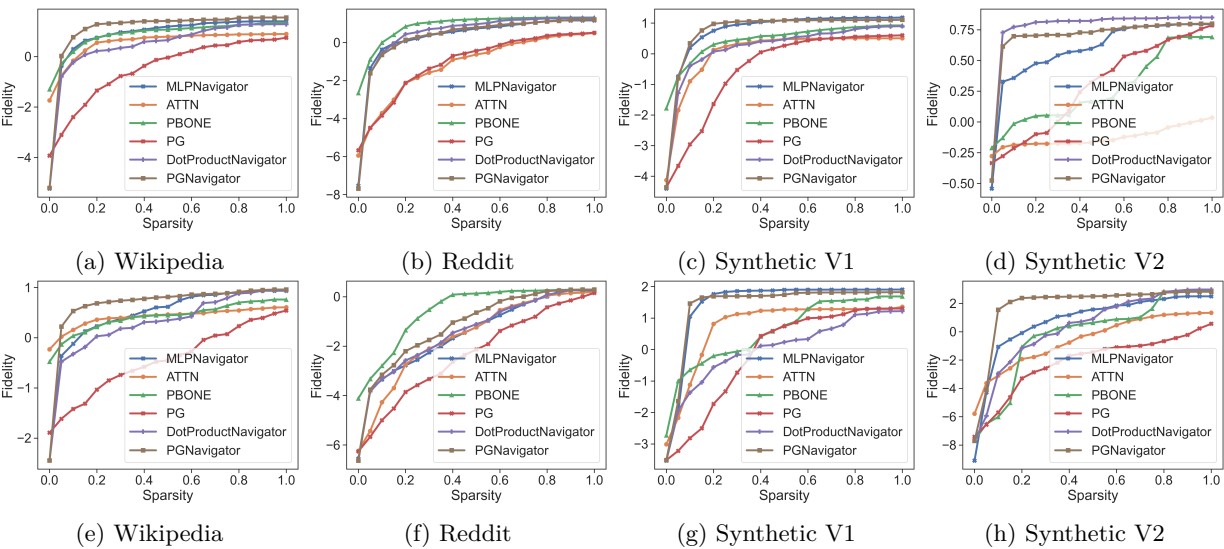

Figure 7: † Fidelity-sparsity curves for TGAT (a-d) and TGN (e-h) at threshold 25.

# B    Model Training

The TGAT and TGN models underwent training using distinct sets of hyperparameters, as anticipated. Additionally, the hyperparameters varied across different datasets. The specific details of these hyperparameters are outlined below.

1. **TGAT** 2 sampled edge neighbours; 2 hidden layers; 2 attention heads; learning rate 0.0001; dropout probability 10%; hidden node embedding dimension 100; hidden time embedding dimension 100.

2. **TGN** 10 sampled edge neighbours; 2 hidden layers; 2 attention heads; learning rate 0.0001; dropout probability 10%; hidden memory dimension 4 for synthetic datasets, 172 otherwise; hidden node embedding dimension 100; hidden time embedding dimension 100; and message dimension 100.

The models for the real datasets were trained with batch size 512 for 10 epochs, while the synthetic datasets trained with batch size 256 for 100 epochs. The Adam optimizer was used for both models.

|  | TGAT | | TGN | |
|---|---|---|---|---|
|  | ACC | AP | ACC | AP |
| Synthetic V1 | 0.9143 | 0.9470 (0.9632) | 0.9352 | 0.9540 (0.9687) |
| Synthetic V2 | 0.8831 | 0.9082 (0.9641) | 0.9436 | 0.9603 (0.9535) |
| Wikipedia | 0.9042 | 0.9724 (0.9791) | 0.8624 | 0.9544 (0.9851) |
| Reddit | 0.9290 | 0.9814 (0.9750) | 0.9252 | 0.9623 (0.9664) |

Table 3: ‡ Initial training results of the models, accuracy (ACC) and average precision (AP). Reported numbers by original authors in brackets.

## C  Original Results and Differences

Furthermore, in Table 4a and 4a we list the full results of the original authors. For clearer comparison between their results and ours, in Table 4b and 5b are listed the relative differences between their reported results and our reproduction.

Table 4: Results comparison with the original study, for the TGAT model.

(a) Results for TGAT model from the original paper.

|  | Wikipedia | | Reddit | | Synthetic V1 | | Synthetic V2 | |
|  | Best FID | AUFSC | Best FID | AUFSC | Best FID | AUFSC | Best FID | AUFSC |
|---|---|---|---|---|---|---|---|---|
| ATTN | 0.891 | 0.564 | 0.658 | -0.654 | 0.555 | 0.390 | 0.605 | 0.291 |
| PBONE | 0.027 | -2.227 | 0.167 | -2.492 | 0.044 | -2.882 | 0.096 | -4.771 |
| PG | 1.354 | 0.692 | 0.804 | -0.369 | 0.476 | -0.081 | 1.329 | -0.926 |
| PGNavigator | 1.836 | 1.477 | 1.518 | 1.076 | 0.780 | 0.666 | 1.630 | 1.331 |

(b) Relative difference in results for TGAT model with respect to the original paper (in %).

|  | Wikipedia | | Reddit | | Synthetic V1 | | Synthetic V2 | |
|  | Best FID | AUFSC | Best FID | AUFSC | Best FID | AUFSC | Best FID | AUFSC |
|---|---|---|---|---|---|---|---|---|
| ATTN | -40.546 | -85.443 | -93.785 | -82.411 | 57.212 | 52.502 | -21.486 | -412.173 |
| PBONE | 3381.139 | -124.126 | 294.332 | -113.917 | 2760.941 | -129.916 | 1176.928 | -118.316 |
| PG | -54.233 | -146.531 | -10.673 | -156.970 | 50.210 | 407.540 | -63.990 | -11.286 |
| PGNavigator | -37.082 | -42.960 | -48.038 | -33.124 | 94.020 | 71.568 | -29.114 | -66.640 |

Table 5: Results comparison with the original study, for the TGN model.

(a) Results for TGN model from the original paper.

|  | Wikipedia | | Reddit | | Synthetic V1 | | Synthetic V2 | |
|  | Best FID | AUFSC | Best FID | AUFSC | Best FID | AUFSC | Best FID | AUFSC |
|---|---|---|---|---|---|---|---|---|
| ATTN | 0.479 | 0.073 | 0.575 | 0.289 | 2.178 | 1.624 | 0.988 | -0.634 |
| PBONE | 0.296 | -0.601 | 0.340 | -0.256 | 0.001 | -3.311 | 0.320 | -5.413 |
| PG | 0.464 | -0.231 | 0.679 | 0.020 | 2.006 | 0.626 | 1.012 | -1.338 |
| PGNavigator | 0.866 | 0.590 | 1.362 | 1.113 | 2.708 | 2.281 | 4.356 | 3.224 |

(b) Relative difference in results for TGN model with respect to the original paper (in %).

|  | Wikipedia | | Reddit | | Synthetic V1 | | Synthetic V2 | |
|  | Best FID | AUFSC | Best FID | AUFSC | Best FID | AUFSC | Best FID | AUFSC |
|---|---|---|---|---|---|---|---|---|
| ATTN | 197.033 | 980.014 | 186.780 | -437.141 | -72.596 | -74.254 | -81.690 | 129.742 |
| PBONE | 466.775 | -224.938 | 778.892 | -153.866 | 73351.165 | -113.057 | -17.116 | -88.612 |
| PG | 184.266 | -95.200 | 45.838 | -11664.245 | -72.589 | -166.895 | -85.158 | 62.869 |
| PGNavigator | 110.238 | 148.700 | 107.432 | 58.986 | -66.004 | -70.174 | -93.906 | -132.758 |

