# OpenReview forum: "[Re] Explaining Temporal Graph Models through an Explorer-Navigator Framework"
_TMLR — Accepted by TMLR_

### Review · Reviewer_tqWE · 2024-02-29

**Summary Of Contributions:**

This framework uses an explorer-navigator method to compute sparse explanations for the predictions made by these complex models. The contributions and new knowledge presented include the following:
1. Reproduction of Main Findings: The authors of the submission have replicated the main findings of the original paper by Xia et al. (2023), confirming the effectiveness of the T-GNNExplainer in providing sparse and interpretable explanations for Temporal GNNs.
2. Extension with Different Navigator Methods: The authors have extended the original work by proposing and evaluating a new type of navigator method, called the DotProductNavigator, intended to enhance the explanation capabilities of the framework.
3. In-depth Examination of Explanation Capabilities and Efficiency: They have performed a detailed analysis of how various model settings and hyperparameters affect the explanation capabilities and computational efficiency of the T-GNNExplainer.
4. Quantitative Analysis of Hyperparameters: The submission includes an analysis of two hyperparameters—namely, the number of rollouts in the Monte Carlo Tree Search (MCTS) and the number of candidate events—on the runtime and fidelity of the explanations.
5. Performance Evaluation: The authors have confirmed that the T-GNNExplainer outperforms other baseline methods across nearly all datasets and evaluation metrics. They also highlight that the navigator helps to bias the search process and that the method can find an exact influential event set.
6. Feasibility Study and Future Work Motivation: The paper concludes with a feasibility study that motivates future research directions in explaining Temporal GNNs.

**Audience:**

Yes

**Claims And Evidence:**

Yes

**Requested Changes:**

1. Clarification of Discrepancies: The authors must clarify the discrepancy between the navigator method described in the original paper and the one implemented in their study. This is critical to ensure the reproduction study's integrity and the comparison's validity.
2. Detailed Implementation Information: The authors should provide detailed descriptions of their implementation, including the algorithms for the new navigator methods, to enable others to replicate their work accurately.
3. Limitations and Failure Cases: The authors should discuss the limitations of their approach and any potential failure cases. This is crucial for readers to understand the boundaries within which the T-GNNExplainer performs effectively.
4. Enhanced Baseline Comparison: Including a wider array of competitive baselines for comparison would enhance the robustness of the performance evaluation.

**Strengths And Weaknesses:**

Strong Aspects:
1. Reproducibility: The submission does an excellent job of reproducing the original paper's results by Xia et al. (2023), which is critical for validating the claims made by the original authors. This strengthens the credibility of the T-GNNExplainer framework.
2. Novel Extensions: The introduction and evaluation of different navigator methods, especially the new DotProductNavigator, provide meaningful additions to the existing literature and could potentially lead to more effective explanation tools for temporal GNNs.
3. Detailed Analysis: The authors' in-depth examination of hyperparameters and their impact on the model's performance offers valuable insights into how the T-GNNExplainer operates and can be optimized for better efficiency and accuracy.

Weaker Elements:
1. Clarity on Navigator Discrepancies: The authors note discrepancies between the navigator method described in the original paper and the provided implementation. It would be beneficial if the authors could offer a more precise explanation of these differences and their implications.
2. Implementation Details: To ensure that others can maintain the reproducibility of the study, the authors should provide comprehensive details about their implementation, including explicit descriptions of the new navigator methods.
3. Baseline Comparison: Although the submission states that T-GNNExplainer outperforms other baselines, the authors should ensure that they have included a diverse range of competitive baselines to benchmark against to rule out any potential biases towards their method.
4. Limitations and Failure Cases: The authors should discuss any limitations of their method, including scenarios where T-GNNExplainer may not perform well.

---

> ### Author Response · Authors · 2024-04-10
> **Response to Reviewer tqWE**
>
> Dear Reviewer,
>
> Thank you for your detailed analysis and valuable insights. In the revised version, we made an effort to address your concerns, which can be summarized as follows:
>
> **(RC1): Clarification of Discrepancies.**
>
> To address this issue, we introduced a new subsection (3.2), which describes how the attention scores are calculated in the TGAT block (Xu et al., 2020) and how the authors of Xia et al., 2023 aggregated these scores to implement the ATTN baseline as well as the PGNavigator. We then made use of these definitions to more rigorously define the different navigators, as well as the ATTN baseline.
>
> **(RC2): Detailed Implementation Information**
>
> We now give equations for each navigator method, which, paired with Appendix A.3 (Changelog) contains all relevant information for implementing these methods from scratch.
>
> **(RC3): Limitations and Failure Cases**
>
> We made improvements to Section 6.1 (Feasibility) to highlight the most crucial issues with the proposed approach, while also highlighting its strengths. We believe this section compactly summarizes all necessary information the reader may need to evaluate whether this approach is applicable to their needs.
>
> **(RC4): Enhanced Baseline Comparison**
>
> There was a similar request in the review notes of Xia et al., 2023, where the authors pointed out that to their knowledge, at the time of writing, there was no other work published that would constitute a good baseline. That is with the exception of the self-interpretable Transformer Hawkes Process (THP) model, which we did not include in our analysis.
> That being said, we found a recent publication, Chen and Ying, 2023, which uses the T-GNNExplainer as a baseline and also significantly outperforms it. In the revised version, we dedicated a paragraph to this in the introduction and also included it in our feasibility study.

---

### Review · Reviewer_pXPi · 2024-03-05

**Summary Of Contributions:**

The paper is a reproducibility study of the **T-GNNExplainer** model proposed in (Xia et al., 2023) at ICLR 2023. T-GNNExplainer adds a trainable "navigator" module on top of a temporal GNN (T-GNN) that predicts the relevance of each possible pair of edges ("events"). Then,it uses this navigator to guide a search process for the most influential set of events that constitute the "explanation" for the prediction.

The authors have re-implemented the codebase from the original paper, highlighting a major inconsistency from the paper (in the navigator part), and comparing three navigators: the one from the original codebase, the one described in the original paper, and another (simpler) variation they propose (a dot-product based navigator). They also evaluate some hyper-parameters that were set as defaults in the original paper, including the number of rollouts in the search algorithm.

I am not very familiar with reproducibility studies, especially in TMLR (is this a special section of the journal?). I'll do my best to evaluate the paper on its own merits, but my evaluation is that the paper is a potential good reproducibility study if the weaknesses below are addressed.

**Audience:**

Yes

**Broader Impact Concerns:**

N/A.

**Claims And Evidence:**

Yes

**Requested Changes:**

R1. I suggest highlighting better the need for a reproducibility study. The authors initially mention that "*The provided repository contained the code for the target models, the T-GNNExplainer model and the baselines, as well as shell scripts and configuration files to run the experiments, train the target models and pre-process the datasets.*", making me wonder why a reproduction was needed. Issues with the code are only mentioned on page 11, with no details on what specific issues were found. I think a short summary of some inconsistencies or code difficulties (e.g., missing requirements) can be added to the introduction.

R2. Concerning the sentence: "*The learned model weights are shared across the authors’ and our navigator implementations, as well as with the PGExplainer baseline.*" Can you clarify what you mean by "*shared across the authors’ implementations*"? Are they taken from the original codebase? Is this pipeline consistent with the original paper?

R3. I suggest adding some equations to clarify the methodology, including (a) the structure of the T-GNN, (b) the specific equations for the navigators, and (c) the optimization task that was used to train the navigator (unless this is common in reproducibility studies, in which case you can ignore this comment).

R4. I also suggest adding a table listing the main reproductions and/or inconsistencies of the paper compared to the original one.

R5. The paper needs a proofediting, especially in the mathematical part. I list three small concerns below as examples.

* p2: $S$ should be $\mathcal{S}$.
* p3: "*the authors define fidelity*", fidelity is a standard metric in explainability literature, I suggest rewording this sentence.
* p4: "$\mathcal{C}(\mathcal{N}^i, e_j)$ should be $\mathcal{C}(\mathcal{N}^i)$.

**Strengths And Weaknesses:**

First, let me consider the need for a reproducibility study. The authors claim at the very beginning that "*Understanding the provided code base proved to be difficult*". I have tried running the original code and I can validate this claim, it has practically no documentation or comments, and installing it runs into many errors due to missing requirements (e.g., dig, tick, ...). By comparison, the codebase provided by the reproducibility study is well documented, clear, with several notebooks and explanations. I think that the need for a reproducibility study can be highlighted more, see **requested changes R1-R2** below.

The paper is clear in the first part, a bit less on the methodological part without reading in detail the original paper. I am not sure what are the guidelines here. In particular:

1. The structure of the encoder-decoder T-GNN is unclear. I am assuming it is using some graph attention mechanism since attention is mentioned in the baselines ("*The computed mean attention scores of the target model*"), but this is unclear.

2. I don't understand how the navigator is trained from this paper. Similarly, I do not understand the differences between the first two navigators they investigate, and how these are related to the "PGExplainer" baseline (which does not look similar to the original PGExplainer paper, to be honest). See **requested change R3** below.

The experimental results are interesting, and I think that the added hyper-parameter evaluations add value to the original paper. There seems to be some super-position between the experiments on the rollout parameter and some results presented by the authors in the rebuttal phase (https://openreview.net/forum?id=BR_ZhvcYbGJ, in particular "Response to Reviewer 1, Q2"). Can the authors clarify this point?

Concerning the comparison with the original paper, the abstract hints that almost all results were correctly reproduced, while instead the experiments hints at several discrepancies ("*Overall, we conclude that the results of the paper are not completely reproducible*"). See **requested change R4** below.

---

> ### Author Response · Authors · 2024-04-10
> **Response to Reviewer pXPi**
>
> Dear Reviewer,
>
> We extend our thanks for the insightful feedback and detailed examination of the submission. We also welcome any further questions and suggestions.
>
> To clear up any possible confusion regarding the intent behind this paper, we are providing the current submission for a special edition of TMLR that focuses on reproducing papers from key conferences and journals in the past year.
>  Addressing your requests:
>
> **R1. Highlighting the need for a reproducibility study.**
>
> We dedicate a section of the paper to the motivation behind this specific study, namely the need for more explainability methods and more explainable models within the emerging applications for TGNNs. To our knowledge, priorly not many papers have been published on this particular topic, which underscores the importance of exploring such methods. Additional clarification behind our motivation has been provided in §1 and §2.
>
> **R2. Clarification behind navigator implementation.**
>
> We have provided detailed equations for each of the proposed navigator methods we explored, as well as further clarification and a diagram for the navigator proposed by the authors. Briefly, the authors propose 3 baselines, AttentionExplainer, PBONE Explainer and PGExplainer (adapted from Luo et al. and augmented with AttentionExplainer). The trained parameters for the baseline PGExplainer were re-used by the navigator for their own TGNNExplainer (more specifically in our case, the PGNavigator).
>
> **R3. Clarification behind methodology.**
>
> As above, we have provided detailed equations behind the underlying explainer baselines and the navigators. Furthermore, we have provided more background on how the target TGNN models operate.
>
> **R4. Addressing discrepancies between original paper and reproduction.**
>
> The original paper produced meaningful results and, overall, our research reproduces some relative results (that is, TGNNExplainer conclusively outperforms the baselines). However, we noted that our results (§5.1) showed that the improvement over the baselines is not as high as the authors claimed - surpassing them by a significant margin. Our results for their proposed explainer were generally a bit lower, even taking into account the high variance in their metrics.
> There is a comparison table in Appendix §C comparing the relative differences between our reproduction and their original results.
>
> **R5. Required proofreading.**
>
> We attempted to correct any issues with the mathematical foundation of the methods and to correct any other mistakes we noticed.

---

### Review · Reviewer_2PhJ · 2024-03-27

**Summary Of Contributions:**

This is an experimental paper that reruns the experimental study of T-GNNExplainer (Xia et al, ICLR23) plus a few additional experiments. I'd hesitate to position this paper as a reproducibility study since it uses the original implementation, the original metrics, the original datasets, original hyperparameters (largely), and even the original anecdotal examples. It does add an ablation of replacing the navigator (one component of the T-GNNExplainer model) by simpler ones as well as on the impact of two hyperparameters.

**Audience:**

No

**Claims And Evidence:**

Yes

**Requested Changes:**

-

**Strengths And Weaknesses:**

S1. Reproduction studies are generally good to have (but see W1)

S2. Highlights a few inconsistencies between T-GNNExplainer paper and the actual implementation

S3. Explores impact of some design choices not explored in original paper

W1. Not really a reproducibility study as it uses the original implementation, the original metrics, the original datasets, original hyperparameters (largely), and even the original anecdotal examples.

W2. Differences in results not clearly explored. The paper points out that their results are somewhat different to the ones in the original paper at times, but does not investigate why this is the case. It's also not clear how much variance there is in these performance numbers, as the authors only do one run on a subset of the test data; e.g., it's net clear whether or not all of the differences are due to variances.

W3. Does not go beyond original work and hence does not provide any tangible and interesting insight. In particular, the paper does not explore the potential and limitations of T-GNNExplainer. It also does not relate it to other work (e.g., Lie et al., "A differential geometric view and explainability of gnn on evolving graphs", also ICLR23).

---

> ### Author Response · Authors · 2024-04-10
> **Response to Reviewer 2PhJ**
>
> Dear Reviewer,
>
> Thank you for your insights. We have revised our submission to the best of our ability, keeping your concerns in mind. As we noted in our reply to Reviewer pXPi, we intend to submit to a special edition of the TMLR journal, explicitly concerned with reproducibility studies. That being said, please see our responses to your valued concerns below.
>
> **(W1):**
>
> We recognise your concerns regarding the reuse of the provided code base. Our intention with this work was twofold. First, we wanted to see if we could replicate the results of Xia et al. ICLR23. Secondly, noting the discrepancies between the reported approach and the provided code base, we performed further experiments, trying to faithfully reproduce the reported architecture and hyperparameter configurations and experiment with several hyperparameters.
>
> **(W2):**
>
> In the revised version, we highlight the differences in the results and provide reasonable grounds for what might be the cause for these. Overall, we partly attribute these differences to the discrepancies we found in the provided code base and try to compensate by introducing the MLPNavigator and using the reported hyper-parameters. In the revised version, we tried to highlight these more clearly.
>
> **(W3):**
>
> Section 6.1 explores the strengths and weaknesses of the proposed approach, highlighting the tradeoff between the increased cost of running this model and the corresponding performance gain. In the revised version we further highlight that the PGNavigator was explicitly tailored to work with the TGAT and TGN target models and that it is not very applicable to other architectures. That being said, in our report, we refrain from making any definitive judgment and instead try to provide the reader with all the necessary details to determine whether the proposed framework is applicable in their context and adapt the explainer framework and methods accordingly.
>
> We additionally compared with the framework and results by Chen and Ying (TempME: Towards the Explainability of Temporal Graph Neural Networks via Motif Discovery, 2023), another paper on the subject of TGNN explainability published post factum.

---

### Decision · Action_Editor_mKxS · 2024-04-21

**Recommendation:** Accept as is

**Comment:**

TMLR encourages submission of reproducibility reports and this paper has done a great job on this, and not only reproduced the original paper, but also provided additional insights.

I'm not recommending a certification as the approach being studied here is not the best performing one and the scope of the additional insight is a bit small.

**Audience:**

Reproducing published work provides additional insight into the existing work. Practitioners will find this work useful. Reviewer pXPi commented on this nicely:

> The authors claim at the very beginning that "Understanding the provided code base proved to be difficult". I have tried running the original code and I can validate this claim, it has practically no documentation or comments, and installing it runs into many errors due to missing requirements (e.g., dig, tick, ...). By comparison, the codebase provided by the reproducibility study is well documented, clear, with several notebooks and explanations. I think that the need for a reproducibility study can be highlighted more

**Claims And Evidence:**

This paper is a reproducibility report. It aims to reproduce the T-GNNExplainer work from Xia et al. 2023, and mainly targets to show (1) the T-GNNExplainer work performs favorably compared to other baselines; (2) the navigators improve performance and (3) the T-GNNExplainer work can produce sparse and accurate interpretations.

This paper also claims a few extensions of the original work, by exploring several different navigator frameworks, and analyzing the impact of a couple important hyper-parameters.

Both reviewers pXPi and tqWE praised the quality of the reproduction. The evidence shows that the authors have managed to reproduce the main claims of the original paper, and also noted a few discrepancies from the original paper. This strengthens the claims of the original paper but also added useful information to the T-GNNExplainer work. The additional navigators and hyperparameter exploration added to the depth of this report. Reviewer pXPi further commented that “As an expert in a related area, I found a few interesting insights reading it.”

Reviewer 2PhJ was concerned that the reproduction was performed with the original authors’ codebase. This is a valid concern but in my opinion even reproducing the results with the original codebase requires an understanding of the work and configuring the codebase to run the experiments. It certainly would be a stronger reproduction if the authors have reimplemented the T-GNNExplainer work from scratch, but I think in its current form the reproduction study still provides useful insights and value.